# Overexpression of Potential Markers of Regulatory and Exhausted CD8^+^ T Cells in the Peripheral Blood Mononuclear Cells of Patients with B-Acute Lymphoblastic Leukemia

**DOI:** 10.3390/ijms24054526

**Published:** 2023-02-24

**Authors:** Mahdieh Naghavi Alhosseini, Marianna Palazzo, Luigi Cari, Simona Ronchetti, Graziella Migliorati, Giuseppe Nocentini

**Affiliations:** Pharmacology Division, Department of Medicine and Surgery, University of Perugia, 06132 Perugia, Italy

**Keywords:** B-acute lymphoblastic leukemia, Treg markers, CD8 exhaustion markers

## Abstract

B-acute lymphoblastic leukemia (B-ALL) is one of the most common pediatric cancers, wherein regulatory T cells (Treg) and exhausted CD8^+^ T cells may be important in its development and maintenance. In this bioinformatics study, we evaluated the expression of 20 Treg/CD8 exhaustion markers and their possible roles in patients with B-ALL. The mRNA expression values of peripheral blood mononuclear cell samples from 25 patients with B-ALL and 93 healthy subjects (HSs) were downloaded from publicly available datasets. Treg/CD8 exhaustion marker expression was normalized with that of the T cell signature and correlated with the expression of Ki-67, regulatory transcription factors (FoxP3, Helios), cytokines (IL-10, TGF-β), CD8^+^ markers (CD8α chain, CD8β chain), and CD8^+^ activation markers (Granzyme B, Granulysin). The mean expression level of 19 Treg/CD8 exhaustion markers was higher in the patients than in the HSs. In patients, the expression of five markers (CD39, CTLA-4, TNFR2, TIGIT, and TIM-3) correlated positively with Ki-67, FoxP3, and IL-10 expression. Moreover, the expression of some of them correlated positively with Helios or TGF-β. Our results suggested that Treg/CD8^+^ T cells expressing CD39, CTLA-4, TNFR2, TIGIT, and TIM-3 favor B-ALL progression, and targeted immunotherapy against these markers could be a promising approach for treating B-ALL.

## 1. Introduction

Following the discovery of thymus-derived CD4^+^ regulatory T cells (tTreg), which are considered CD25^+^FoxP3^+^CD127^−/low^ [1,2,3,4], several studies demonstrated that murine and human Tregs are crucial for maintaining immune homeostasis. Therefore, Treg dysfunction might determine autoimmune disease occurrence and favor tumor growth. However, the role of Tregs in malignancies is complex and context-dependent [5,6]. Recent advances in Treg knowledge have resulted in the discovery of various CD4^+^ T cell lineage subtypes, several of which are termed peripherally derived Treg subsets (pTreg) because they differentiate in the periphery [1,7,8]. Both tTreg and pTreg subsets express Treg markers, including CTLA-4, GITR, 4-1BB, ICOS, and TIGIT [9,10]. At least one tTreg subset expresses not only FoxP3, but also Helios, another transcription factor [11,12], and some pTreg subsets do not express FoxP3 [13,14,15,16].

CD8^+^ T cell exhaustion has been demonstrated as a dysfunction mode of T cells determined by Tregs and other regulatory cells and favoring tolerance of tumors in the tumor microenvironment (TME) [17]. Most exhaustion markers (e.g., PD-1 and TIM-3) are also Treg markers [17,18,19]. Currently, modulation of exhausted T cells is used in several anticancer protocols, and modulation of Treg is in the evaluation as anticancer treatment [10,18,20]. However, considering the phenotypic heterogeneity of Treg and exhausted T cell populations, identifying the most effective target in each tumor is crucial for designing successful immunotherapeutics in the future.

B-acute lymphoblastic leukemia (B-ALL) is one of the most common pediatric cancers, accounting for 85% of all ALL cases. B-ALL typically affects children aged <6 years but is also diagnosed in older children and adult populations with an incidence of approximately 1–5 per 100,000 persons per year worldwide [21]. Despite genetic alterations in B-ALL having long been proven as the basis of the disease [22], much data suggests that infection may be responsible for the disease [23,24], indicating that immune system cells, including Tregs, are key in B-ALL development and maintenance [25,26].

Leukemias have unique features in comparison to solid tumors and the role of Tregs is more complex in hematological than non-hematological cancers [27,28]. Some studies examined whether the tTreg number in CD4^+^ cells present in the peripheral blood mononuclear cells (PBMC) is modulated in patients with ALL, including B-ALL, compared to that of healthy subjects (HSs) [25,29,30,31]. Overall, these studies reported that patients with B-ALL had increased percentages of CD25^+^FoxP3^+^ and CD25^+^CD127^−/low^ tTregs but had decreased percentages of CD25^+^ cells [25,29,30]. Despite some studies correlating Treg frequency with immunotherapy efficacy [32,33], tTregs were associated with either poor prognosis or increased survival [28,34].

In the present study, we used bioinformatics to evaluate the potential role of 20 Treg/CD8 exhaustion markers in patients with B-ALL. Specifically, we evaluated their expression in patients with B-ALL and HSs and correlated marker expression with that of cytokines, transcription factors, and a cell proliferation marker. The results demonstrated the existence of five Treg/CD8 exhaustion markers that might be treatment targets in patients with B-ALL.

## 2. Results

### 2.1. Treg/CD8 Exhaustion Marker Overexpression in Patients with B-ALL

We evaluated Treg and exhausted CD8^+^ T cell presence in patients with B-ALL by evaluating Treg/CD8 exhaustion marker expression in PBMC via expression data from bioinformatics datasets. Considering that most PBMC in patients with B-ALL are leukemic blasts and that gene expression is evaluated in PBMC, we first evaluated the T cell levels in the PBMC of 93 HSs and 25 patients with B-ALL using a recently established T cell signature (TCS) [13]. The signature does not indicate the number of T cells present in a sample but provides quantitative information on the level of T cell-derived mRNA, enabling comparison among samples [13]. As expected, the patients’ TCS mRNA values were ~4 log2 lower than that of the HSs (Appendix A), suggesting that the mean mRNA level from the patients’ PBMC T cells was approximately 16-fold lower than that in the HSs.

The expression of 20 Treg and/or exhausted CD8^+^ T cell markers was evaluated to assess the presence of Treg subsets and/or exhausted CD8^+^ T cells in the PBMC of patients and HSs. The mRNA level of the markers in each sample was normalized with that of the TCS to estimate the marker expression level within the PBMC T cell population. With the exception of FCRL3, the mean value of normalized marker expression in the patients’ PBMC was greater than that in the HSs’ PBMC (Figure 1A). Figure 1B shows the inter-patient variation of marker overexpression that is relevant, especially for some markers.

### 2.2. Correlation of Treg/CD8 Exhaustion Marker Overexpression with Higher Leukemic Cell Proliferation Rate

Higher Treg and exhausted CD8^+^ cell numbers are associated with more advanced solid and leukemic tumors, where removing or targeting these cells can restore the anti-tumor activity of the immune system, which suppresses tumor progression [35,36,37]. Therefore, we investigated whether Treg/CD8 exhaustion marker expression levels in PBMC correlated with increased leukemic cell proliferation by evaluating Ki-67 expression.

Assuming that the main proportion of Ki-67 expression in PBMC relies on leukemic cells, because they comprise the majority of the sample and are the only proliferating cells, we tested whether Ki-67 expression correlated with Treg/CD8 exhaustion marker expression normalized by TCS (i.e., the percentage of Treg subsets and/or exhausted CD8^+^ T cells within the T cells). Five markers (CD39, CTLA-4, TNFR2, TIGIT, and TIM-3) were significantly positively correlated with Ki-67 and showed correlation coefficient values >0.60 (Figure 2 and Figure 3). The data represented in Figure 3 were cleaned from three outliers. The positive correlation between the expression of five markers with Ki-67 was present even before cleaning the data from outliers (Appendix A).

CD39, CTLA-4, TIM-3, and TNFR2 were overexpressed by 2.3–3.8-fold, and TIGIT was overexpressed by 6.3-fold in the patients’ T cells (Appendix A).

### 2.3. Treg Phenotypes of Cells Expressing Treg/CD8 Exhaustion Markers

To evaluate whether cells with regulatory function express the markers that were expanded in the patients and that favored leukemic growth, we tested whether the marker expression correlated positively with Treg transcription factor and suppressive cytokine expression. The expression of four of the five selected markers reported above correlated positively with that of the FoxP3 transcription factor (Table 1), which was overexpressed in the patients’ PBMC (Appendix A).

Therefore, the data supported the hypothesis that FoxP3^+^ Tregs express CD39, CTLA-4, TIGIT, and TIM-3. TIGIT expression correlated positively not only with FoxP3 expression, but also with that of the Helios transcription factor (Table 1, Figure 4), which is expressed by a tTreg subset [11,38].

The expression of the five markers correlated with that of IL-10 (Table 1), the suppressive cytokine secreted by tTreg and pTreg. Therefore, the data supported the hypothesis that these cells produce IL-10. Moreover, TNFR2 and TIM-3 expression correlated with TGF-β expression. Both IL-10 and TGF-β were overexpressed in the patients by >20-fold (Appendix A).

The findings might suggest that in patients with leukemia: (1) TIGIT is expressed by at Treg subset that is FoxP3^+^Helios^+^ and expresses IL-10; (2) CTLA-4 and CD39 are expressed by tTreg and/or pTreg subsets that are FoxP3^+^Helios^−^ and express IL-10; (3) TIM-3 is expressed by a tTreg and/or pTreg subset that is FoxP3^+^Helios^−^ and expresses both IL-10 and TGF-β; (4) TNFR2 is expressed by a FoxP3^−^Helios^−^ pTreg subset that expresses IL-10 and TGF-β.

### 2.4. Treg/CD8 Exhaustion Marker Expression by Exhausted CD8^+^ T Cells

We also investigated if CD8^+^ T cells expressed CD39, CTLA-4, TNFR2, TIGIT, and TIM-3. To this end, we evaluated whether CD8α chain and CD8β chain expression correlated with that of the five markers (for the correlation, we considered the CD8α and CD8β chain mean expression). The expression of all five markers was significantly positively correlated with CD8 chain expression (Table 2). Moreover, both chains were overexpressed in the patients as compared to the HSs (Figure 5A,B).

As TIM-3, CD39, CTLA-4, and TIGIT are well-known CD8^+^ exhausted T cell markers [39,40], it is reasonable to suggest that CD8^+^ T cells expressing these markers would produce low levels of proteins involved in the effector function of activated CD8^+^ T cells, for example, Granzyme B and Granulysin. That is, a patient with high levels of TIM-3, CD39, CTLA-4, and TIGIT would demonstrate low levels of Granzyme B and Granulysin, and vice versa. Indeed, TIM-3, CD39, CTLA-4, and TIGIT (but not TNFR2) correlated negatively with Granzyme B and Granulysin (Table 2). Contrary to the CD8α and CD8β chains, Granzyme B and Granulysin expression levels were not different between the patients and controls (Figure 6A,B), suggesting that CD8^+^ T cells increased in the patients, but the active cells [i.e., cytotoxic T lymphocytes (CTL)] expressing Granzyme B and Granulysin were not.

To examine if there was a relationship between Treg/CD8 exhaustion marker expression and exhausted CD8^+^ T cells, we subtracted the CD8^+^ expression marker levels (the mean expression of CD8α and CD8β chains) from the CTL marker expression levels (the mean of Granzyme B and Granulysin) to evaluate the percentage of exhausted CD8^+^ T cells among the CD8^+^ T cells. Interestingly, the positive correlation between the exhausted CD8^+^ T cells and TIM-3, CD39, CTLA-4, and TIGIT expression yielded a higher R-value than that observed when evaluating the correlation between the selected markers and the whole CD8^+^ T cell population, further suggesting that TIM-3, CD39, CTLA-4, and TIGIT are also markers of exhausted CD8^+^ T cells. On the contrary, TNFR2 was not correlated with the latter analysis, suggesting that it is not a marker of exhausted CD8^+^ T cells, which was in agreement with previous findings [41,42].

## 3. Discussion

In this study, we analyzed 20 genes mainly expressed in Treg and/or exhausted CD8^+^ T cells and identified five markers (CD39, CTLA-4, TIGIT, TIM-3, and TNFR2) overexpressed in the PBMC of pediatric patients with B-ALL. The increased expression of the proliferation marker Ki-67 in the patients with a high level of these markers suggested that these markers potentially identify cells favoring leukemic cell growth (Figure 2). Ki-67 expression correlates with uncontrolled proliferation rate, worse prognosis, and worse treatment response in hematological malignancies, including ALL [43,44]. Therefore, the correlation of CD39, CTLA-4, TIGIT, TIM-3, and TNFR2 with Ki-67 suggested that their overexpression in Treg and/or exhausted CD8^+^ T cells of patients with ALL favors leukemia development and treatment resistance. As far as we know, no study described results suggesting which marker is useful to identify T cells favoring leukemic growth.

Studies from the last 15 years evaluated the Treg percentage in pediatric B-ALL. Bhattacharya et al. reported that patients with leukemia had fewer CD25^+^ T cells among the CD4^+^ cells as compared to HSs, but that patients’ peripheral T cells demonstrated increased FoxP3, IL-10, and TGF-β levels [29]. On the contrary, two studies reported increased CD4^+^CD25^+^Foxp3^+^ Tregs in the PBMC of patients with B-ALL compared to that of HSs, while Liu et al. confirmed increased IL-10 and TGF-β levels in such patients [25,30,45]. Interestingly, our findings were in line with their results, demonstrating increased FoxP3, CD25, IL-10, and TGF-β in patients with B-ALL. However, we demonstrated that CD25 expression did not correlate with Ki-67 expression, suggesting that CD25^+^ cells do not favor leukemia cell proliferation, probably because CD25 is expressed not only by Tregs but also by activated conventional CD4^+^ and CD8^+^ T cells.

Among the overexpressed markers favoring leukemic cell growth, we found that TIGIT positively correlated with Helios and FoxP3. Helios^+^ Tregs demonstrated stronger suppressive capacity in vitro [46]. Moreover, patients with B-ALL have a higher percentage of Helios^+^FoxP3^+^CD4^+^ Tregs, and Helios expression was positively correlated with the Treg suppressive function in patients with B-ALL [47]. Our findings suggest that in patients with B-ALL, Helios^+^FoxP3^+^CD4^+^ Tregs express TIGIT and favor leukemic growth. The finding is original but in line with previous studies.

Several studies suggested that pTregs favor both solid and hematological cancer growth independently of CD25 and FoxP3 expression. For example, CD39^+^ Tregs in gastric and colon cancer were identified as a potentially important immunosuppressive subset [48,49], and a higher number of TIGIT^+^Foxp3^+^ γδ T cells was associated with poor overall survival in patients with acute myeloid leukemia (AML) [50].

Accordingly, we studied the expression of several Treg markers expressed by both tTregs and pTregs. Only Liu et al. evaluated three of the same markers in children with B-ALL, demonstrating that the patients’ CTLA-4, GITR, and LAG3 mRNA levels were higher than those in HSs [30]. Moreover, Zhang et al. reported increased TIM-3 expression on CD4^+^ cells in post-allogenic hematopoietic stem cell transplantation (HSCT) B-ALL relapse [51]. Our study confirms that the PBMC of patients with B-ALL had increased CTLA-4, LAG3, GITR, and TIM-3 expression, but suggests that only the cells overexpressing CTLA-4 and TIM-3 favor leukemic growth.

The demonstration that Tregs expressing CD39, TIGIT, and TNFR2 are expanded in the PBMC of patients with B-ALL is original. Moreover, we suggest also that not only the cells that overexpress CTLA-4 and TIM-3, but even those that overexpress CD39, TIGIT, and TNFR2, favor B-ALL leukemia growth. The expansion of Tregs in the PBMC of patients with B-ALL is expected. In fact, several pieces of evidence suggest that Tregs infiltrate the TME and favor the growth of solid cancers determining a suppressive microenvironment [52]. If we consider bone marrow and blood as the main microenvironment of leukemic cells, it is not surprising to find expanded Tregs in the PBMC of patients with leukemia, as demonstrated for patients with AML [52,53]. Some years ago, our group demonstrated that the phenotype of TME-infiltrating Tregs is peculiar for each tumor [13] and, in our opinion, the reasons are to be found in the ligands expressed by cancer cells and the cytokine and chemokines secreted by the TME-infiltrating cells, peculiar for each cancer. Recently, B-ALL-derived exosomes have been demonstrated to regulate immune function in human T cells [54]. Our finding might suggest that one or more Treg subsets expressing CTLA-4, TIM-3, CD39, TIGIT, and TNFR2 are expanded and relevant for the development and expansion of B-ALL cells.

The finding, if confirmed, may be crucial in setting up new anti-leukemic treatments. In fact, killing such cells by antibodies favoring antibody-dependent cell-mediated cytotoxicity (ADCC) and complement-dependent cytotoxicity (CDC) may favor the immune response against leukemic cells. Recently published studies demonstrate the efficacy of such a treatment type in patients with multiple myeloma [55,56].

Similar to chronic infection, T cells in the TME develop the exhausted phenotype and function. Exhausted T cells in cancer express high levels of inhibitory receptors, including PD-1, CTLA-4, TIM-3, LAG3, BTLA, and TIGIT, most of which are also known as Treg markers. Consistent with the expression of these inhibitory markers, T cells demonstrate impaired production of effector cytokines, such as IL-2, TNF-α, IFN-γ, and Granzyme B [4]. In many mouse models and human solid tumors, including melanoma, ovarian cancer, and non–small cell lung carcinoma, CD8^+^ T cells demonstrate T cell exhaustion and dysfunction [20]. TIM-3, TIGIT, and CTLA-4 upregulation in CD8^+^ T cells with exhaustion features has been reported in chronic lymphocytic leukemia (CLL), AML, and post-allogenic HSCT B-ALL relapse [51,57,58], and targeting these exhaustion markers improved the outcome of hematological malignancies [59].

Our study demonstrates that TIM-3, TIGIT, and CTLA-4, overexpressed in the patients with B-ALL, correlate not only with transcriptional factors and cytokines expressed by Tregs, but also with CD8α and β chain^+^ T cells, suggesting that these markers are expressed not only by CD4^+^ Treg, but also by CD8^+^ T cells. More interestingly, TIM-3, TIGIT, and CTLA-4 expression correlated negatively with Granzyme B and Granulysin expression, suggesting that the increased expression of these markers is accompanied by lower Granzyme B and Granulysin expression by CD8^+^ cells (i.e., CD8^+^ cells are exhausted). As far as we know, no study has described TIM-3, TIGIT, and CTLA-4 as CD8^+^ exhaustion markers in the PBMC of patients with B-ALL.

Although CD39 is not considered an exhaustion marker, CD39 overexpression by CD8^+^ cells in different tumors often signifies exhaustion [60]. Additionally, CD39, which has nucleotidase activity, is upregulated in CD8^+^ T cells in response to cytokines during tolerance induction in vivo and is accompanied by IL-12 and IL-10 secretion [61]. Our findings were in line with the findings of that aforementioned study, as we demonstrated that patients with B-ALL have: (1) IL-10 overexpression; (2) positive correlation of CD39 expression with CD8 chains expression; and (3) negative correlation with Granzyme B and Granulysin. Interestingly, CD39 has been considered a useful marker to monitor B-ALL [62].

TNFR2 is not an exhaustion marker. Indeed, TNFR2 activation causes NF-κB activation and cell growth in some cell types, including activated T lymphocytes [63]. Accordingly, TNFR2 expression was not negatively correlated with Granzyme B and Granulysin expression.

The strengths of our study are as follows: (1) knowing that marker expression may differ between children and adults, we evaluated data only from pediatric patients and HSs (0–18 years old); (2) we evaluated the expression of several Treg/CD8 exhaustion markers in PBMC from an appropriate number of patients (*n* = 25) and HSs (*n* = 93); (3) our analysis focused not only on tTreg markers, but also on pTreg markers; (4) to best of our knowledge, this is the first study that performs bioinformatics analysis of CD8^+^ exhaustion marker levels in the PBMC of patients with pediatric B-ALL.

The main limitation of our study is the use of data evaluating the mRNA levels of the genes, as the mRNA expression level of a gene does not always correlate with the expression level of the coded protein. Another limitation of this study is that bioinformatics analysis of mRNA levels does not establish whether the overexpressed genes are expressed only by Treg, CD8^+^ exhausted T cells, or other cells. We attempted to resolve the issue by performing the correlations shown and discussed earlier and not by studying the genes possibly expressed by leukemic blasts.

## 4. Materials and Methods

### 4.1. Data Source, Tumor Type, and Controls

We obtained gene expression data from the publicly available datasets in Gene Expression Omnibus [64] and European Bioinformatics Institute [65] repositories. The normalized absolute gene expression values generated using the Affymetrix Human Genome U133 Plus 2.0 platform were downloaded using the Genevestigator V3 suite (NEBION AG, Zurich, Switzerland). Using normalized data enables comparison within individual datasets and among datasets derived from the same tissue. The search keyword was “B-acute lymphoblastic leukemia”. The data selection criteria were: (1) gene expression was evaluated with microarray mRNA expression profiling; (2) the organism was *Homo sapiens*; (3) each dataset included PBMC samples. Based on the above criteria, we extracted the gene expression data from the PBMC samples of 25 patients with B-ALL and 93 HSs. The data were obtained from pediatric patients and HSs (0–18 years old) and are expressed as log2 values.

### 4.2. Genes of Interest

In this bioinformatics study, we evaluated the expression of genes coding 20 Treg markers [13]. In particular, we examined the expression of the following genes: *TNFRSF9* (4-1BB), *CCR4*, *CCR8*, *IL-2RA*, *ENTPD1* (CD39), *CD226*, *CTLA-4*, *CXCR3*, *FCRL3*, *TNFRSF18* (GITR), *HLA-DRA*, *ICOS*, *IL-21R*, *LAG3*, *TNFRSF4* (OX40), *PDCD1* (PD-1), *PDCD1LG2* (PD-L2), *TIGIT*, *HAVCR2* (TIM-3), and *TNFRSF1B* (TNFR2). To gain a deeper understanding of subset characterization, the expression levels of *FOXP3* (FoxP3), *IKZF2* (Helios), *IL10* (IL-10), *TGFB1* (TGFβ), and the proliferation marker *MKI67* (Ki-67) were correlated to the markers. Moreover, we evaluated *CD8A* (CD8α chain), *CD8B* (CD8β chain), *GZMB* (Granzyme B), and *GNLY* (Granulysin) expression to examine CD8^+^ cell marker expression. All raw expression data used in the study are reported in Appendix A.

### 4.3. Use of T Cell Signature to Normalize Gene Expression

We modified a previously published T cell signature (TCS) [66] to evaluate the level of T cells present in the HS and patient PBMC. As B-ALL aberrant blasts can express T cell-specific genes (including those included in the TCS), this could have led to an inaccurate evaluation of the patients’ T cell population levels. Therefore, we tested each gene in the TCS and excluded the genes that were outliers. Specifically, we evaluated the expression level (expressed as the log2) of each gene in the TCS of the PBMC from the HSs (mean HS) and patients (mean PT) and recorded the difference in expression (mean HS − mean PT = Δmean). The mean (overall Δ mean) and standard deviation (SD) of all TCS genes were calculated and the genes with a Δ mean higher or lower than the overall Δ mean ± SD were excluded. Then, a second round of the same procedure was run. Consequently, six genes were excluded from the TCS: *CD3D*, *CD6*, *TIGIT*, *PHYIN1*, *ITK*, and *ICOS*. Therefore, the modified TCS used in this study included nine genes: *CD2*, *CD247*, *CD28*, *CD3G*, *GPR171*, *GZMK*, *KLRB1*, *TRAT1*, and *TRBC1*/*TRBV25*-*1*.

### 4.4. Statistical Analysis

All statistical analyses were performed using Prism 8 software (GraphPad Software, San Diego, CA, USA). Normality was tested using the Kolmogorov–Smirnov (KS) test. Differences between the two groups were compared using an independent sample *t*-test (KS test passed) or Mann–Whitney test (KS test failed). The correlation of any Treg marker expression with other markers was analyzed with the Spearman (KS test failed) or Pearson (KS test passed) correlation tests. *p*-values < 0.05 was considered significant.

## 5. Conclusions

Our results indicated the existence of five surface proteins (CD39, CTLA-4, TIGIT, TIM-3, and TNFR2) overexpressed by Treg and/or exhausted CD8^+^ T cells in peripheral T lymphocytes that apparently favor leukemic cell growth. While some of our conclusions confirmed the data of other studies, which indicated the possible success of our strategy, experimental studies on the peripheral CD4^+^ and CD8^+^ T cells of pediatric patients with B-ALL should be undertaken to verify these findings. In this context, our study indicated the markers that should be prioritized in future experimental studies. If the data are confirmed, antibodies targeting these markers might be useful for treating patients with B-ALL.

## Figures and Tables

**Figure 1 ijms-24-04526-f001:**
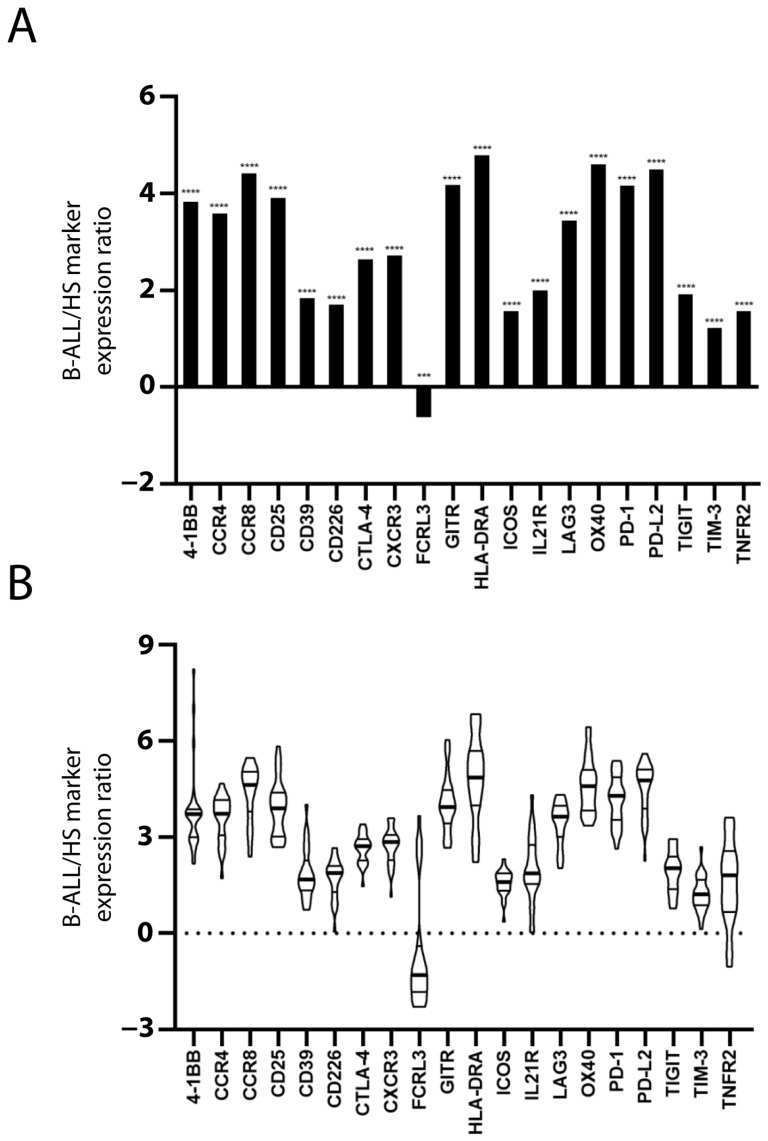
**Increased Treg/CD8 exhaustion marker expression in patients with B-ALL.** Treg/CD8 exhaustion marker expression (log2) in patient and healthy subject (HS) PBMC was normalized with the T cell signature (TCS) expression levels (log2). The mean normalized marker expression of patients minus HSs is reported (log2) (**A**). The significant difference between patients and HSs was evaluated using the *t*-test and Mann–Whitney test. *** *p* < 0.001 and **** *p* < 0.0001. The inter-patient variation of Treg/CD8 exhaustion marker overexpression is represented as violin plot (**B**). The distribution of data extends above the largest value and extends below the smallest value. The median (thicker line), and the 3rd and 1st quartiles are shown within the box.

**Figure 2 ijms-24-04526-f002:**
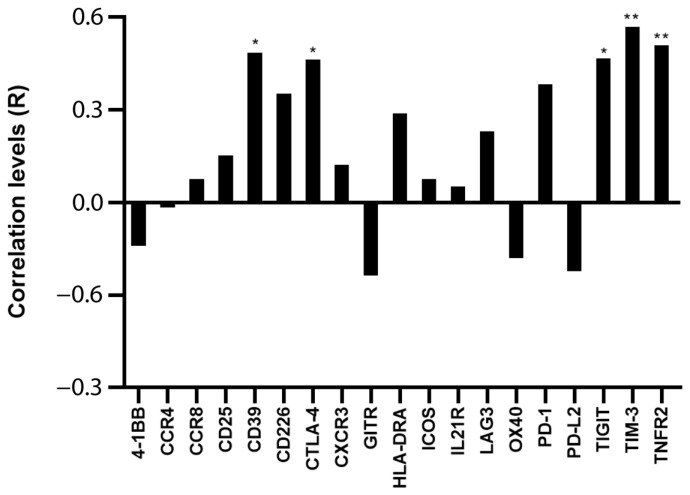
**Correlation between Ki-67 and Treg/CD8 exhaustion markers.** The rho factor of correlation (R) between mRNA expression of Ki-67 and normalized mRNA expression of Treg/CD8 exhaustion markers is reported. Data are evaluated without cleaning any outlier. The correlation significance was evaluated using Pearson and Spearman tests. * *p* < 0.05, ** *p* < 0.01.

**Figure 3 ijms-24-04526-f003:**
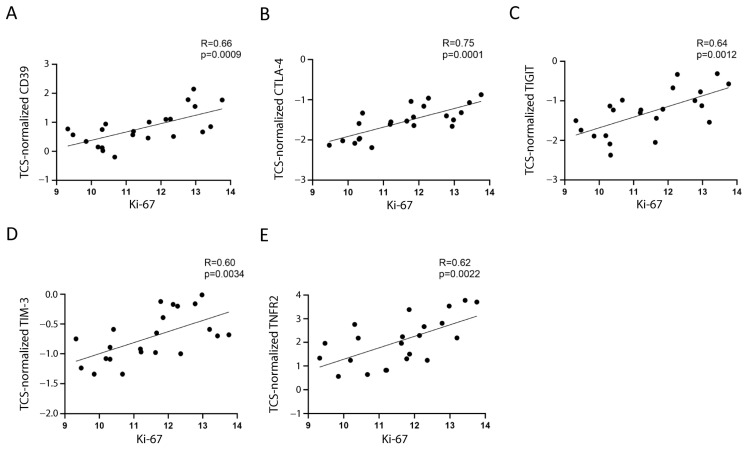
**Correlation between Ki-67 and the selected markers**. The correlation between mRNA expression of Ki-67 and normalized mRNA expression of CD39 (**A**), CTLA-4 (**B**), TIGIT (**C**), TIM-3 (**D**), and TNFR2 (**E**) is shown. The three values furthest from the line of regression were considered to be outliers and were deleted in each figure; the correlations including the outliers are shown in Appendix A. The rho factor of correlation (R) and significance evaluated using Pearson and Spearman tests are reported.

**Figure 4 ijms-24-04526-f004:**
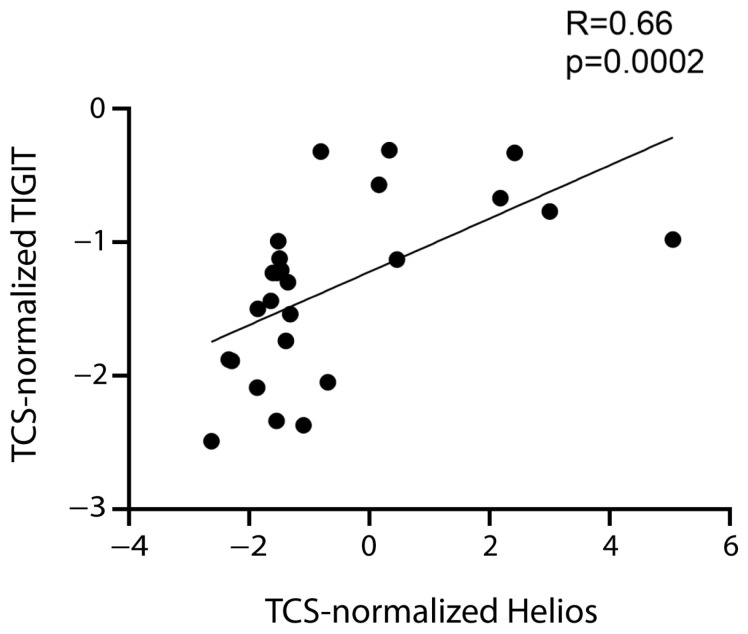
**Correlation between TIGIT and Helios.** Correlation between normalized mRNA expression (log2) of TIGIT and Helios. The rho factor of correlation (R) and significance evaluated using the Spearman test are reported.

**Figure 5 ijms-24-04526-f005:**
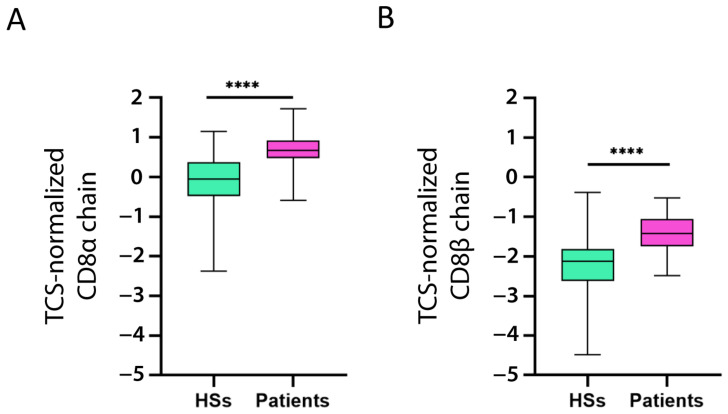
**Increased CD8 chain expression in patients with B-ALL.** CD8α chain (**A**) and CD8β chain (**B**) mRNA expression (log2) in patient and healthy subject (HS) PBMC was normalized with the T cell signature (TCS) expression levels (log2). The data are represented as box-and-whisker plots. The box limits represent the 1st and 3rd quartiles. The median is represented as a line within the box. The whiskers go down to the smallest and up to the largest value. The significant difference between patients and HSs was evaluated using the *t*-test. **** *p* < 0.0001.

**Figure 6 ijms-24-04526-f006:**
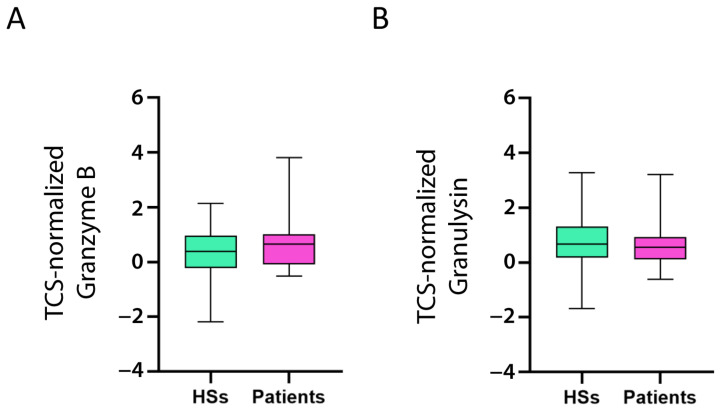
**Expression of activated CD8^+^ T cell markers in HSs and patients with B-ALL.** Granzyme B (**A**) and Granulysin (**B**) mRNA expression (log2) in patient and healthy subject (HS) PBMC was normalized with the T cell signature (TCS) expression levels (log2). The box limits represent the 1st and 3rd quartiles. The median is represented as a line within the box. The whiskers go down to the smallest and up to the largest value. The significant difference between patients and HSs was evaluated using the *t*-test and Mann–Whitney test. Expression values do not differ.

**Table 1 ijms-24-04526-t001:** Correlation (R) between normalized mRNA expression of surface Treg markers and that of Treg phenotypic/functional genes.

	CD39	CTLA-4	TIGIT	TIM-3	TNFR2
**FoxP3**	0.66	0.73	0.86	0.56	ns
**Helios**	ns	ns	0.66	ns	ns
**IL-10**	0.62	0.47	0.52	0.51	0.43
**TGF-β**	ns	ns	ns	0.40	0.66

ns, not-significant correlation.

**Table 2 ijms-24-04526-t002:** Correlation (R) between normalized mRNA expression of selected genes and CD8 markers.

	TIM-3	CD39	TIGIT	CTLA-4	TNFR2
**mean of CD8α chain + CD8β chain**	0.50	0.50	0.56	0.61	0.58
**mean of Granzyme B + Granulysin**	−0.55	−0.58	−0.76	−0.58	ns
**(mean of CD8α chain + CD8β chain)—(mean of Granzyme B + Granulysin)**	0.68	0.67	0.81	0.73	ns

ns, not-significant correlation.

## Data Availability

Not applicable.

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
