# Peer review of "Overexpression of Potential Markers of Regulatory and Exhausted CD8^+^ T Cells in the Peripheral Blood Mononuclear Cells of Patients with B-Acute Lymphoblastic Leukemia"

_ijms, 2023, doi:10.3390/ijms24054526_

Round 1

Reviewer 1 Report

In this manuscript, the authors identified by a bioinformatic study a panel of genes potentially involved in B-ALL progression. Although preliminary, the data they presented are well-organized and discussed, and help providing new pharmacological targets to be explored.

I have only minor suggestions to improve the quality of the manuscript.

1. The correlation data without outliers are presented in the supplementary. I would rather present these data in the main text, specifying that they are "cleaned" from outliers, and put the whole panel of data in the supplementary files. 

2. I would not divide the discussion into titled paragraphs, but organize it in an organic way.

Author Response

Comment 1: The correlation data without outliers are presented in the supplementary. I would rather present these data in the main text, specifying that they are "cleaned" from outliers, and put the whole panel of data in the supplementary files. 

Response: The correlation data without outliers was presented in the main text in Figure 3 and it was specified that the data was cleaned from outliers (lines 124-125). The data containing the outliers were represented in Figure S2.

Comment 2: I would not divide the discussion into titled paragraphs, but organize it in an organic way.

Response: The discussion was reorganized in the organic style as the Reviewer suggested.

Reviewer 2 Report

This manuscript was clear, relevant for the field and presented in a well-structured manner. The authors studied the mRNA expression values of peripheral blood mononuclear cell samples from B-ALL patients and healthy subjects from publicly available datasets. The authors reached a conclusion that Treg/CD8+ T cells expressing CD39, CTLA-4, TNFR2, TIGIT, and TIM-3 may favor B-ALL progression. Such discovery suggests that targeted immunotherapy against these markers could be a promising approach for treating B-ALL.

The authors provided sufficient background information in the introduction part. The authors discovered the overexpression of Treg/CD8 exhaustion marker in patients with B-ALL. Also, there is correlation of those marker overexpression with higher leukemic cell proliferation rate. In addition, the authors took a deeper look into Tregs and CD8+ T cells. In conclusion ,the authors found the existence of CD39, CTLA-4, TIGIT, TIM-3, TNFR2 overexpressed by Treg and/or exhausted CD8+ T cells in peripheral T cells that favors leukemic cell growth. The authors also pointed out the markers that should be prioritzed in future experimental studies. Also, such discovery can help future B-ALL therapy development.

However, it would be better if the authors can discuss the potential mechanism for this discovery. 

The language of the manuscript is clear and professional. But some improvement can still be made in some places.

Y axis label for figure 1 only mentioned Treg, but in the title of the figure, the authors used "Treg/CD8 exhaustion". It would be better to use the same wording. It would also be better if the authors can show the inter-patient variation of some markers.

In figure 3, it is a little bit confusing that the authors used p=* in the figure and p<=0.05 in the caption. It is not so clear whether p=0.05 or smaller than 0.05. Similar problem in figure 4.

The font of the some words in line 80 and 81 seems different from others. 

Author Response

Comment 1: In conclusion, the authors found the existence of CD39, CTLA-4, TIGIT, TIM-3, and TNFR2 overexpressed by Treg and/or exhausted CD8+ T cells in peripheral T cells that favors leukemic cell growth. The authors also pointed out the markers that should be prioritized in future experimental studies. Also, such discovery can help future B-ALL therapy development. However, it would be better if the authors can discuss the potential mechanism for this discovery.

Response: We have discussed the potential mechanism(s) in lines 248-265 of the revised manuscript.

Comment 2: The language of the manuscript is clear and professional. But some improvement can still be made in some places.

Response: The whole manuscript has been checked, some sentences have been rewritten and the level of English has been improved.

Comment 3: Y-axis label for figure 1 only mentioned Treg, but in the title of the figure, the authors used "Treg/CD8 exhaustion". It would be better to use the same wording.

Response: As correctly pointed out by the Reviewer, the Y-axis label was edited.

Comment 4: It would also be better if the authors can show the inter-patient variation of some markers.

Response: The inter-patient variation of all markers was elaborated and represented in Figure 1B.

Comment 5: In figure 3, it is a little bit confusing that the authors used p=* in the figure and p<=0.05 in the caption. It is not so clear whether p=0.05 or smaller than 0.05. Similar problem in figure 4.

Response: As truly mentioned by the Reviewer, the problem with p values was corrected in figures and legends. The p values were written in figures and the range of p-value in legends was removed to avoid the confusion.

Comment 6: The font of the some words in line 80 and 81 seems different from others.

Response: The font size was corrected.